# Combined FCS and PCH Analysis to Quantify Protein Dimerization in Living Cells

**DOI:** 10.3390/ijms22147300

**Published:** 2021-07-07

**Authors:** Laura M. Nederveen-Schippers, Pragya Pathak, Ineke Keizer-Gunnink, Adrie H. Westphal, Peter J. M. van Haastert, Jan Willem Borst, Arjan Kortholt, Victor Skakun

**Affiliations:** 1Department of Cell Biochemistry, University of Groningen, 9747 AG Groningen, The Netherlands; l.m.schippers@rug.nl (L.M.N.-S.); p.pathak@rug.nl (P.P.); a.keizer-gunnink@rug.nl (I.K.-G.); p.j.m.van.haastert@rug.nl (P.J.M.v.H.); 2Laboratory of Biochemistry, Wageningen University & Research, 6708 WE Wageningen, The Netherlands; adrie.westphal@wur.nl (A.H.W.); JanWillem.Borst@wur.nl (J.W.B.); 3Department of Systems Analysis and Computer Simulation, Belarusian State University, 220030 Minsk, Belarus

**Keywords:** brightness and diffusion global analysis, *Dictyostelium discoideum*, dimeric protein, GFP, FK506 binding protein 12, fluorescence correlation spectroscopy, fluorescence fluctuation spectroscopy, photon counting histogram

## Abstract

Protein dimerization plays a crucial role in the regulation of numerous biological processes. However, detecting protein dimers in a cellular environment is still a challenge. Here we present a methodology to measure the extent of dimerization of GFP-tagged proteins in living cells, using a combination of fluorescence correlation spectroscopy (FCS) and photon counting histogram (PCH) analysis of single-color fluorescence fluctuation data. We named this analysis method brightness and diffusion global analysis (BDGA) and adapted it for biological purposes. Using cell lysates containing different ratios of GFP and tandem-dimer GFP (diGFP), we show that the average brightness per particle is proportional to the fraction of dimer present. We further adapted this methodology for its application in living cells, and we were able to distinguish GFP, diGFP, as well as ligand-induced dimerization of FKBP12 (FK506 binding protein 12)-GFP. While other analysis methods have only sporadically been used to study dimerization in living cells and may be prone to errors, this paper provides a robust approach for the investigation of any cytosolic protein using single-color fluorescence fluctuation spectroscopy.

## 1. Introduction

Oligomerization and complex formation of proteins in organisms is one of the key mechanisms for the orchestration of protein function and activities. Classic examples are the dimerization and activation of protein kinases like extracellular signal-regulated kinase 2 (ERK2) and rapidly accelerated fibrosarcoma (RAF) [1,2,3]. Most nuclear receptors are monomeric in the cytosol and become dimeric and active upon the binding of ligand molecules [4,5], although, for the human androgen receptor, this may be the other way around [6]. Unraveling the dimerization mechanism of the androgen receptor is important because it has been implicated in hormone-related diseases [7]. In fact, dimerization and oligomerization play a role in many diseases, especially in neurodegenerative diseases, e.g., leucine-rich repeat kinase 2 and alpha-synuclein in Parkinson’s disease and tau protein in Alzheimer’s disease. 

To study the dynamic process of dimerization it is logical to use in cellulo techniques such as ‘live cell imaging’. Over the past decades, numerous approaches have been developed, all based on the non-invasive visualization of fluorescently-labeled proteins. For the detection of dimers it is possible to use a combination of two differently colored labels, as used in techniques like Förster resonance energy transfer (FRET) and fluorescence cross-correlation spectroscopy (FCCS) [8,9]. However, the creation of protein constructs with the right proximity and spectral properties of fluorophores is often a daunting task. Therefore, several techniques have been developed using a single fluorophore, most of which are based on fluorescence fluctuation spectroscopy (FFS).

FFS exploits the fluctuation of fluorescence signals, caused by either diffusion of fluorescent particles, through a tiny, illuminated volume or reversible change of its spectroscopic properties, e.g., singlet-state–triplet-state transitions, recorded from a stack of images or from a time-observed single spot. This fluctuation pattern is influenced by the diffusion speed and brightness of the particles. The diffusion and brightness per particle are different for monomer and dimer because of their different size or mass and number of fluorophores. In this way co-diffusion of multiple proteins in a dimer/oligomer form can be observed without any requirements for proximity of fluorophores within the complex, and monomer can be distinguished from the dimer with the same fluorophore.

To calculate the diffusion and brightness per particle from FFS data, various analysis techniques have been developed [10,11,12]. Some of these techniques have been applied for analysis of the dimerization of GFP-tagged proteins in living cells, i.e., number and brightness analysis (N&B [13]), raster image correlation spectroscopy (RICS [14]), photon counting histogram (PCH [15]) fluorescence intensity distribution analysis (FIDA [16]), and time integrated fluorescence cumulant analysis (TIFCA [12]).

N&B and RICS are based on the analysis of images, yielding valuable information on the location of oligomeric complexes in the cell, especially on membranes. However, the vast number of pixels analyzed simultaneously is at the expense of precision. For an accurate determination of molecular brightness and diffusion parameters in the intracellular environment, single-point FFS as in FCS, PC(M)H, and FI(M)DA would be the method of choice [17,18,19,20,21,22,23].

In single-point FFS, the fluorescence intensity fluctuations are caused by the diffusion of proteins through a very small volume, which is generated by a tightly focused laser beam of a confocal microscope. The fluorescence intensity at a specific time point and the next time point are correlated, and the amount of correlation of a specific time-step is dependent on the diffusion speed of the molecules crossing the observation volume. The correlation shows a time-dependent decay and can be fitted to a chosen autocorrelation function (ACF) describing the physical model of diffusion.

In fluorescence correlation spectroscopy (FCS), these ACF curves are used to estimate the diffusion time and number of fluorescent particles. By simply comparing the latter to the total photon count rate, the molecular brightness can be calculated, which is defined as the average number of photons detected per molecule per second (cpms). FCS has been widely used to study protein concentration, diffusion, kinetics, and aggregation in living cells, as well as dimerization [24,25]. However, to study dimerization in living cells, a more sophisticated and accurate method is required to calculate the molecular brightness such as PCH or FIDA analysis. 

PCH and FIDA are two related techniques, both based on the analysis of the amplitudes in the fluctuation of fluorescence intensity. The distributions of these amplitudes are plotted as photon counting distributions (PCD) and are derived from the same FFS data as the ACF curves. In the case of analysis of multiple distributions, they are called PCMH and FIMDA [18,22,23]. PCMH and FIMDA allow to extract information not only about brightness, but also about diffusion parameters. However, to get the same information content about the diffusion as available in the analysis of a single ACF in FCS, the number of analyzed PCDs should be equal to the number of points in the ACF, which is around one hundred and fifty. The collecting and analysis of a high number of PCDs, calculated at different binning times, is time consuming, whereas the same information, specifically on diffusion, is obtained by fitting just one ACF in FCS. For the determination of brightness, however, PCH and FIDA, and their extensions PCMH and FIMDA, are suitable methods. PCH was first applied to dimerization in cells by Chen et al. using tandem-dimer EGFP as well as concentration-dependent dimerization of several nuclear receptors [15]. For samples with low signal-to-noise ratios, which is often the case in cellular samples, additional analysis tools are required. 

Here we describe the combination of FCS and PCH in the analysis of molecular brightness and diffusion speed and its application in the analysis of dimerization of cytosolic proteins in living cells. A specific analysis technique has been developed which allows the simultaneous analysis of FCS and PCH by linking common parameters [26]. The advantage of this combined FCS and PCH global analysis method is that it enhances the advantages of PCH in estimation of brightness by adding the power of FCS in estimation of diffusion to the analysis [26]. Compared to individual FCS and PCH analyses (as well as other techniques), the combined global FCS and PCH analysis is superior at lower signal-to-noise ratios, making the analysis more robust and accurate. Moreover, this technique can be performed on a standard FCS microscope using single-color fluorescence.

Previously, the combined FCS and PCH global analysis method has been applied to purified dimeric GFP (diGFP) in vitro [26], as well as monomeric GFP in cells [27]. However, the detection and quantification of dimeric species in living cells is a challenging task, and the accuracy of this global analysis, as for all FFS methods, depends on the method of application. We have optimized the combined FCS and PCH global analysis method to detect the state of dimerization of fluorescent molecules in living cells. This combined analysis will be referred here as brightness and diffusion global analysis (BDGA). We use *Dictyostelium* cells as a model system, expressing either monomeric GFP, diGFP, or FKBP12-GFP, a small protein that dimerizes upon ligand binding [28,29], and demonstrate that this BDGA methodology is a suitable approach to study the in vivo dimerization of GFP-tagged proteins.

## 2. Results and Discussion

### 2.1. Optimization of the BDGA Method in Order to Measure Monomer-Dimer Equilibria

In cells, dimeric proteins often exist in an equilibrium between monomers and dimers; however, resolving monomer-dimer mixtures using FFS methods is still a challenging task. Both species differ in mass and brightness by just a factor of two, and even when we applied the standard BDGA method, see details in the Methods section, it was not possible to clearly resolve a monomer-dimer mixture at the low signal-to-noise ratios typical for FFS measurements in cells. Therefore, we aimed to optimize BDGA for a multi-species application in cells. 

To be able to quantify protein dimerization in cells, the method was first validated on data obtained from mixes of cell lysates expressing monomeric GFP and tandem-dimer GFP (diGFP), see Materials section for details. In this way, various stable monomer-dimer equilibria were mimicked in a biological context. The measurements of these samples were performed at typical settings for standard FCS measurements, as described in the Methods section. The total averaged fluorescence intensity ranged from 40 kHz to 250 kHz, depending on the (fluorescent) protein concentration. 

#### 2.1.1. Initial Steps in the Analysis of FFS Data

For the data analysis, raw data sets of photon arrival times were converted into autocorrelation curves (ACF) and photon counting distributions (PCD) (Figure 1, top panels). In order to do so, a set of data was imported into the measurement database of the FFS data processor (FFS DP) and each measurement was split into traces of 5 s. A 5-s trace-length provides enough data points for validated samplings, while yielding a high enough number of traces in the global analysis. Per trace one ACF and three PCDs were generated. The binning time step of the ACF curves was set at 2 × 10^−7^ s, roughly 500 times shorter than the residence time of GFP and diGFP in the confocal volume, which was in the order of 100 µs. The binning times of the PCD curves were set such that the histograms contained a minimum of five data points and the histogram peak value did not pass the 10-photon counts position. The first constraint is required because the PCH model has five parameters for an analysis with two components and the latter to maintain the super-Poissonian shape of the distribution (i.e., to avoid effect of averaging). 

Next, all generated ACF and PCD curves of one measurement were imported in the analysis platform of FFS DP to perform the global analysis. The FCS 3D free diffusion model with triplet state term and brightness correction (Equation (1)) was used to fit ACFs, and the PCH model with out of focus (first order), diffusion and triplet correction (Equations (2)–(9)) was used to fit PCDs. A detector dead-time correction was included for the analysis of measurements obtained at high intensities. These equations can be used for one, two, or more component analyses. We started with a two-component analysis because of our goal to analyze samples that contain a mixture of two species. 

#### 2.1.2. Two-Component BDGA of Lysate Mixtures

In the simplified system of artificial monomer-dimer equilibria in cell lysates, we expected that the parameters of the two species could be resolved via a two-component analysis. Initially, the diffusion parameter *τ_diff_*
_1_, see description of model parameters in Section 3.1 and Appendix A, was grouped, as well as the triplet state parameters *τ_trip_* and *F_trip_* (*τ_diff_*
_2_ were not grouped to allow some freedom in their estimation). Reliable values for these parameters were retrieved from the separate FCS analyses and used to fix *τ_diff_*
_1_, *τ_trip_*, and *F_trip_* in the combined BDGA analysis. As a result, parameters *N*_1_ and *N*_2_ should indicate the amount of monomeric and dimeric particles in the confocal volume, and the brightness parameters *q*_1_ and *q*_2_ should differ by a factor of two. However, the values for *N*_1_ and *N*_2_ varied a lot between different traces within one measurement, while *q*_1_ and *q*_2_ yielded similar values, in-between the brightness of monomeric and dimeric controls (Appendix A), indicating that the two species were not resolved properly. 

In order to obtain good fits, we applied different approaches in the global analysis, mostly including variation in the different parameters to be fixed or free. In the example of a lysate mix with 50% GFP and 50% diGFP, best results were obtained when diffusion parameters *τ_diff_*
_1_ and *τ_diff_*
_2_ were fixed to the values found for GFP (*τ_diff_*
_1_) and diGFP (*τ_diff_*
_2_), while *q*_1_ was fixed to the brightness of GFP (*q*_GFP_) and the *q*_2_/*q*_1_ ratio to *r* = 1.8 (Table 1, Appendix A). Alternative approaches, like fixing *q*_2_ to the brightness found for diGFP, resulted in *N*_1_- and *N*_2_-values that deviated more from the expected percentages for GFP and diGFP (Appendix A). However, the optimal value for *r* varied for measurements with other proportions of diGFP (Appendix A), thus a generally applicable fixed value of *r*, which would yield the expected *N*-values under different conditions, could not be determined. Therefore, the two-component analysis is considered to not be a reliable method for analysis of our measurements of a mix of two components that vary only two-fold in mass and brightness. Moreover, this method is not suitable for unknown samples, since *τ_diff_* and *q* were fixed to known values from the pure samples, and in biological situations it is difficult to obtain proteins that are completely monomer or completely dimer. We concluded that most probably the signal-to-noise ratio needed to resolve such mixtures into monomer and dimer was not reached.

Although the two-component model could not resolve two species which differ by only a factor of two in brightness and mass, in more complex samples the addition of a second component may very well be required. In our experience, adherence to cellular structures may cause part of the particles to diffuse very slowly (in the seconds time-scale), requiring the two-component model for a good fit. But for the lysate mixes, which do not have such a slow-diffusing component, we changed to a simpler variant of the BDGA method, namely the one-component analysis.

#### 2.1.3. One-Component BDGA of Lysate Mixes

While in a two-component analysis the number of molecules (*N*_1_ and *N*_2_) reflects the proportion of each species, in a one-component analysis the parameter of interest is the average brightness *q*. This apparent brightness per particle increases with a higher dimer to monomer ratio. The BDGA analysis was performed as described above, except that a one-component model was applied. *F_trip_*, *τ_trip_*, and *τ_diff_*
_1_ (now called *τ_diff_*) were first calculated in the FCS global analysis, and subsequently fixed in the BDGA analysis. An example of the fit of one of the analyses is shown in Figure 1. The brightness parameter *q* was determined per trace and corrected for correction factor *Fc*_1_ (Equation (12)) before further calculations were made. Data from multiple measurement days were normalized and averaged as described in the Methods section. Of note: the GFP lysate was used for normalization rather than the R110 calibration dye, because it corrects for day-to-day variation between the biological samples rather than only the technical setup.

Brightness per sample was plotted as a ratio compared to the brightness of GFP (Figure 2 and Table 2). Despite some deviation due to technical fluctuation per day, a linear trend is observed with higher average brightness when more dimer is present (solid line). This is slightly lower than the theoretical value when diGFP would be twice as bright as the monomer (dashed line). This deviation is in accordance with previous reports, where a ratio of 1.7 was observed, possibly due to fluorescence energy transfer [26]. A proportion of dimer as small as 5% could be distinguished (*p* < 0.001). Therefore, one-component BDGA can be used to determine whether dimeric proteins are present in the sample and to estimate the monomer to dimer ratio.

### 2.2. Improved BDGA Methodology for the Analysis of Cellular Data

Analysis of cellular data brings new challenges compared to data from cell lysates. Due to internal movement of non-fluorescent biomolecules or compartments in living cells, such as lysosomes or vacuoles, the fluorescence intensity in the small FCS spot will fluctuate over time (Figure 3, top panel). Therefore, each measurement was split into 3-s traces, and instable traces were excluded from further analysis based on their typical aberrant tail in the ACF. This approach has proven to be effective to analyze the non-stationary data [30]. Of note, by using 3- rather than 5-s traces (as for lysate data), less data had to be removed per aberrant trace, because a deviation lasted usually for about three seconds and the selection of fluctuating regions in the measurement became more precise. 

Other challenges to overcome are phototoxicity and photobleaching. Focusing the laser beam on a single spot in a cell for a few minutes will cause local stress such as phototoxicity to the cells. Moreover, photobleaching of the fluorophore can play a role, because if a dimer of which one molecule is bleached it will behave as dimer in FCS and as monomer in PCH. In the small, confined volume of a cell, the bleached molecules are not diluted by diffusion as in lysates. Therefore, *Dictyostelium* cells were measured for only 45 s. All other steps in the ACF and PCD curve generation were the same as above. The resulting ACF as well as the PCD curves from a series of traces did overlay but deviated due to bleaching (Figure 3).

As with the lysate data, *τ_diff_*
_1_, *τ_trip_*, and *F_trip_* were first calculated via the global analysis of only ACF curves. Importantly, *τ_trip_* and *F_trip_* were the same for all samples with single GFP and GFP-fusion proteins, while diGFP had significantly different triplet state values (Appendix A). Therefore, *τ_trip_* and *F_trip_* were averaged over a large number of cells, and all samples except diGFP were analyzed with these values fixed for *τ_trip_* and *F_trip_* in both the FCS and BDGA analyses. Fixation of triplet parameters improved the analysis, yielding lower standard deviations for the brightness parameters than when these parameters were left free (Appendix A). In samples with complex proteins, the standard deviations will be higher and the reduction in standard deviation upon fixing the triplet state parameters is expected to be clearer. More details about the triplet state in FCS can be found in [31,32]. In general, fixing some parameters to the previously known values improved the estimation of any other parameters of interest. 

Representative examples of the global analysis of individual cells are shown in Table 3, while the averages of multiple cells are shown in Table 4 (two first rows). The fit per cell yielded in many cases a χ^2^ close to unity and the standard deviations of fit parameters were similar to what was obtained for measurements in cell lysate (Table 1). However, the variation between cells was much larger compared to the cell lysate measurements. Therefore, 27–35 cells from three measurement days were analyzed per experimental condition, normalized towards GFP and averaged for the parameters *q* and *τ_diff_* (Table 4).

Unlike with cell lysate, in cells the additional recalculation of the apparent brightness to *q_true_* was not performed. The required correction parameter *Fc*_1_ varied to a great extent and did not reflect the real instrumental correction factor due to ‘absorption’ of any deviations which are not accounted by the model and which are more prevalent in cells. Since all measurements were performed on the same instrument with similar settings, a similar *Fc*_1_ correction parameter was expected for the whole data set. Any deviations caused by different *Fc*_1_ should be averaged out because of the large sample size and linearity of the correction. Finally, the influence of the correction parameter *Fc*_1_ to the relative brightness should be canceled out due to the performed normalization. In order to validate the results, a pilot of seven cells per sample was analyzed using PCH with the polynomial model (instead of Gaussian), which does not include parameter *Fc*_1_. Indeed, the differences in relative brightness were small (Appendix A). Therefore, we are confident that the additional calculation of *q_true_* was not required for the comparison of relative values of cellular samples as was performed on the lysate samples. 

The results of the analysis of GFP and diGFP in cells are shown in Table 4 (upper rows). The diffusion of both GFP and diGFP was slower in cells and had a broader distribution compared to cell lysate (Table 2). This broad deviation in diffusion may be due to variation in viscosity, depending on the position in the cell, which could not be distinguished in our setup. Still, the diffusion of GFP and diGFP was significantly different in the separate samples (*p* < 0.001; Table 4), as determined in the initial analysis step with FCS without PCH. 

Considering the significant difference in diffusion between GFP and diGFP, the question may arise why FCS should not be used. However, if a protein with unknown dimerization property is observed, the amount of dimer cannot be estimated because a reference with pure monomer of the same protein is often lacking. Moreover, the diffusion of proteins in cells is often influenced by adhesion to cellular structures like membranes and protein complexes. Nevertheless, FCS may be used to estimate the protein concentration *N*, from which the average brightness per particle may be calculated by dividing the total fluorescence intensity by *N* (*q* = <*I*>/*N*). In complex samples which are difficult to fit, this may be a fast way to assess whether higher order oligomers are present in the cells, since this average brightness can be compared to the brightness of monomeric GFP in cells. This simplification comes at the expense of any corrections for background fluorescence, protein aggregates and other artifacts. 

With the brightness and diffusion global analysis as performed here, the brightness of GFP and diGFP could clearly be distinguished with a significance of *p* < 0.001 (Table 4 and Figure 4, first rows/bars). The magnitude of this difference is 1.6-fold, which is comparable to what was found before with purified protein (1.72; [26]), but smaller than the two-fold difference which is expected when two fluorophores are present per particle. The low apparent brightness of the dimer may be caused by resonance energy transfer, triplet state formation, photobleaching, or blinking of one of the subunits [33,34,35]. In the small confined volume of cells, the effect of photobleaching is substantial, which can be avoided by adjusting the laser power or it may be reduced by using short measurement times (<1 min) as well as more photostable fluorophores or two-photon excitation [15]. However, we strived for a simple, broadly applicable method which could be performed with any existing GFP construct on any confocal microscope with FCS module. Using diGFP as a reference for the brightness of dimeric samples, it will be possible to estimate the amount of dimer in the sample. Taken together, with the above data we have shown that in our setup the BDGA method can distinguish GFP from diGFP in *Dictyostelium* cells.

### 2.3. Application of the Developed Methodology to the Analysis of Induced Dimerization in Cells

To mimic real dimerization and create a monomer-dimer mix in living cells, GFP was fused to the FKBP12 dimerization domain [29,36]. Two of these domains can dimerize via a linker molecule called B/B homodimerizer (or ‘dimerizer’, Appendix A). Measurement and analysis of *Dictyostelium* cells with FKBP12-GFP were performed as described in the Methods section. The ACF and PCD curves of FKBP12-GFP appear similar to those of GFP in cells (Figure 5 compared to Figure 3). 

When dimerizer was added to the cells, the apparent brightness of the FKBP12-GFP molecules was 1.33-fold higher than GFP, which was significantly higher compared to FKBP12-GFP without dimerizer or the GFP controls (*p* < 0.001), indicating that dimerization of FKBP12-GFP was taking place. Considering the linear relationship between average brightness and the monomer-dimer ratio (Figure 2), and the fact that 100% diGFP was 1.6-fold brighter than GFP (Table 4), a brightness of 1.33 × *q*_GFP_ would correspond to approximately 55% dimer. Taken together, the amount of dimeric protein could be estimated in a cellular context using monomeric and dimeric GFP as a reference. 

The difference in diffusion between FKBP12-GFP and ‘FKBP12-GFP plus dimerizer’ is smaller than the difference between GFP and diGFP, but due to a large sample size this difference is still significant (*p* < 0.001). Thus, the presence of FKBP12 dimer can already be distinguished by analyzing diffusion alone. However, as mentioned above, ‘clean’ diffusion (without adherence to other cellular components) and the availability of a 100% monomeric sample, are required to determine whether a sample would contain oligomers based on diffusion. 

Taken together, we show that BDGA is an adequate analysis procedure, even in an in cellulo situation, to quantify the dimerization status of cytosolic proteins with statistical significance. To our knowledge, this is the first time that this method is applied to dimerizing proteins in living cells.

## 3. Materials and Methods

### 3.1. Theory of the Global Analysis of ACF and PCD

The set of measured autocorrelation functions (ACF) and photon counting distributions (PCD) is analyzed globally using a combination of two models, FCS and PCH, by linking parameters having the same meaning (and name) in both models [26]. The analysis is performed using the constrained nonlinear iterative least-squares method with the Levenberg–Marquardt optimization [37]. Application of a global analysis approach increases the information content available from a single measurement that results in more accurate values of molecular diffusion coefficients and triplet-state parameters, and also in robust, time-independent estimates of molecular brightness and number of molecules. 

The FCS model used for fitting the autocorrelation function describes a number of independent molecular species, which diffuse freely in a 3D Gaussian-shaped observation volume and undergo the triplet process, and is written as [38]:(1)$$\begin{array}{c} G(t)=G_{\inf}+\frac{1-F_{trip}+F_{trip}e^{-t/\tau_{trip}}}{(1-F_{trip}){\left(\sum_{i}q_{0eff\;i}N_{0eff\;i}\right)}^{2}}\\ \;\;\;\times \;\sum_{i}\frac{q_{0eff\;i}^{2}N_{0eff\;i}}{\left(1+t/\tau_{diff\;i}\right)\;\sqrt{\left(1+t/a^{2}\tau_{diff\;i}\right)}}, \end{array}$$
where $G_{\inf}=G_{\mathrm{corr}}(\infty )$; *F_trip_* and *τ_trip_* are, respectively, the fraction and the relaxation time of molecules in the triplet state; *q*_0*eff i*_ is the apparent molecular brightness of species *i*, expressed in counts per molecule per second (cpms); *N*_0*eff i*_ is the number of molecules of species *i* in the effective volume *V_eff_*; $a=\omega_{z}/\omega_{xy}$, *ω_xy_* and *ω_z_* are, respectively, the lateral and axial radii of the confocal detection volume; and *τ_diff i_* is the lateral diffusion time of species *i*, which is related to the diffusion coefficient *D* via *τ_diff_* = *ω_xy_*^2^/(4*D*). The effective volume is calculated as $V_{eff}=\chi_{1}^{2}/\chi_{2}$, where $\chi_{k}=\int_{V}B^{k}(r)dV$ and *B*(**r**) is the brightness profile function, which is the convolution of excitation intensity and detection efficiency profiles, depending on the radius *r* of the overlapping excitation and detection spots. The subscript 0 in *q*_0*eff i*_
*and N*_0*eff i*_ means that these parameters do not depend on time. Equation (1) has been written in assumption that each molecular species has the same triplet-state characteristics. 

The PCH model with triplet, free 3D diffusion and out of focus emission corrections is calculated by a numerical algorithm consisting of the following steps (the full protocol is described in details in [38]):Calculate the time dependent parameters
(2)$$q_{\;eff\;i}(T)=q_{\;0\;eff\;i}B_{2}(T),\;N_{\;eff\;i}(T)=N_{\;0\;eff\;i}/B_{2}(T)$$
for each molecular species *i* = 1, 2, …, where *T* is the counting time interval (bin time), *B*_2_(*T*) is the binning correction factor
(3)$$B_{2}(T)=\frac{2}{T^{2}}\int_{0}^{T}(T-t)g(t)dt$$
calculated over a time dependent term of the autocorrelation function in FCS
(4)$$g(t)=\frac{1-F_{trip}+F_{trip}e^{-t/\tau_{trip}}}{\left(1-F_{trip}\right)\left(1+t/\tau_{diff\;i}\right)\;\sqrt{\left(1+t/a^{2}\tau_{diff\;i}\right)}}$$
and *q*_0*eff*_ *_i_*, *N*_0*eff*_ *_i_*, *a*, *F_trip_*, *τ_trip_*, *τ_diff_ _i_* are fit parameters;Calculate single-molecular PCD $p^{\left(1\right)}(k,q_{\;eff\;i})$ for each molecular species *i*
(5)$$\begin{array}{l} p_{}^{\left(1\right)}(0,q_{eff\;i})=1-\sum_{k=1}^{K}p_{}^{\left(1\right)}(k,q_{eff\;i}),\;\\ p^{\left(1\right)}(k,q_{eff\;i})=\frac{1+Fc_{2}}{{\left(1+Fc_{1}\right)}^{2}}\left[p_{3DG}^{\left(1\right)}(k,q_{eff\;i})+\frac{1}{k!\Theta }\sum_{n=k}^{\infty }\frac{{\left(-1\right)}^{n-k}{\left(q_{eff\;i}T\right)}^{n}Fc_{n}}{(n-k)!{\left(2n\right)}^{3/2}}\right], \end{array}$$
where *k* = 1, 2, …, *K* is the number of photons detected in an interval *T*, *K* is the maximal number of photons, *Fc_n_*, *n* = 1, 2 are instrumental out-of-focus correction parameters (*Fc_n_* are also fit parameters) and
(6)$$p_{3DG}^{\left(1\right)}(k,q_{eff\;i})=\frac{1}{\Theta \sqrt{\pi }k}\int_{0}^{\;\infty }\gamma (k,q_{eff\;i}Te^{-x^{2}})dx.$$In Equation (6), *γ*() is the incomplete gamma function and parameter Θ is varied depending on the value of the product of *q_eff_ T* (from 1 to 20), see details in [38];Calculate PCD *P*(*k*) for each brightness component assuming the Poissonian distribution of a number of molecules in an open observation volume
(7)$$P(k,q_{eff\;i},N_{eff\;i})=\sum_{M=0}^{\infty }p^{\left(M\right)}(k,q_{eff\;i})\;Poi(M,QN_{eff\;i}),$$
where $p^{\left(M\right)}(k,q_{eff\;i})=\underset{M\;times}{\underbrace{p^{\left(1\right)}(k,q_{eff\;i})⊗\ldots ⊗p^{\left(1\right)}}}(k,q_{eff\;i})\;$ is M-times convolution of the single-molecule PCD and *Poi*(*k*,*η*) denotes the Poisson distribution with the mean value *η*; Calculate the total PCD for a molecular system. PCD of a number of independent species is given by a convolution of PCD of each species
(8)$$P(k)=P(k,N_{eff1},q_{eff1})⊗\ldots ⊗P(k,N_{eff\;n},q_{eff\;n})$$The correction on dead-time is performed accordingly to the following equation [39]:(9)$$P_{DT\;corr}(k)=\sum_{j=0}^{\infty }P_{0}(k+j)P_{binomial}\left(j;k+j,\frac{(k+j)\tau_{dt}}{T+(k+j)\tau_{dt}}\right)$$
where *τ_dt_* is the detector dead time (fit parameter) and *P_binomial_*(*j, n, p*) is the binomial probability distribution. We omitted the dependence on *T* from *q_eff_ _i_*, *N_eff_ _i_*, and therefore from all related expressions in the Equations (5)–(9) for the sake of simplicity.

Both FCS and PCH models can be modified to fit the brightness ratio $r_{i}=q_{0eff\;i}/q_{0eff\;1}$ instead of fitting of the absolute brightness values, thus, to have the following set of fitting parameters: *q*_0*eff* 1_, *r*_2_, *r*_3_, …. The ratio can be then fixed to the expected value or constrained to reasonable boundaries, which increases the ability of the model to resolve multicomponent samples. The FCS model will take the form:(10)$$\begin{array}{c} G(t)=G_{\inf}+\frac{1-F_{trip}+F_{trip}e^{-t/\tau_{trip}}}{(1-F_{trip}){\left(\sum_{i}r_{i}N_{0eff\;i}\right)}^{2}}\\ \;\;\;\times \;\sum_{i}\frac{r_{i}^{2}N_{0eff\;i}}{\left(1+t/\tau_{diff\;i}\right)\;\sqrt{\left(1+t/a^{2}\tau_{diff\;i}\right)}}. \end{array}$$

In the PCH model, one has to recalculate the time dependent parameters $q_{eff\;i}(T)$ accordingly:(11)$$q_{eff\;i}(T)=r_{i}\;q_{0eff\;1}B_{2}(T).$$

For the sake of simplicity, *q* will be used instead of $q_{0eff}$, *N* instead of $N_{0eff}$, and *r* instead of *r*_2_ throughout the text.

In the described method the FCS and PCH models are linked through the common parameter *N*. Values of *N* in FCS are corrected by values of *q* from PCH, and values of *q* and *N* in PCH are corrected by values of diffusion and triplet state parameters from FCS. When using the two-component model to resolve two species in the sample, it is important that the two fractions are separated in the same way for each trace in the global analysis of multiple traces, especially at low signal-to-noise ratios and/or when *q* and *N* values are close for the two species. To prevent the components to be exchanged in some individual traces, e.g., *q*_1_ of that trace is low and *q*_2_ is high while for the rest of the traces it is the other way around, the allowed range of each component should either be linked across all traces or even better, be tightly constrained or fixed. The swapping of components during the fit process (and therefore trapping in the local minima) is probable, even inside a global analysis of one ACF and one PCD in the case when brightness and diffusion are close to each other, which is typical for the dimerization studies. If necessary, the swapped values can be manually corrected and the analysis should then be repeated. The use of the model with brightness ratio fixed to the expected value may drastically improve the resolvability of the method in this case. 

### 3.2. Sample Preparation and Measurement

#### 3.2.1. Plasmids and Cloning

Vectors pDM317 and pDM334 [40] were used for constitutive or inducible expression of GFP-S65T (further referred to as GFP), respectively. For the construction of tandem-dimer GFP (diGFP), an additional GFP gene was added to these plasmids by digesting GFP from pDM313 with SpeI/XbaI and ligating into the SpeI site of pDM317 or pDM334. This created the following linker: 5′-SGLRSSTSS-3′ between the two GFP inserts. 

FKBP12 was cloned into pDM334 using BglII/SpeI restriction sites. To improve binding of the dimerizing ligand, an F36V point mutation was created by PCR-based site directed mutagenesis using Phusion (ThermoScientific) polymerase and primer: 5′-GATGGAAAGAAAGTTGATTCCTCCC resulting in FKBP12 F36V (hereafter referred to as FKBP12). 

#### 3.2.2. Cell Culture

The axenic *Dictyostelium discoideum* wild-type strain was used for all experiments. Cells were grown in HL5-C medium (Formedium) at 22 °C. The indicated constructs were transformed in AX2 cells by electroporation and selected with either 10 µg/mL geneticin or 50 µg/mL hygromycin B. To induce expression from the tetracycline inducible vectors, the cells were overnight incubated with 1 µg/mL doxycycline. Dimerization of FKBP12-GFP was induced by incubation with B/B homodimerizer (Clontech, also called AP20187) to a final concentration of 1 µM for 3 h. 

#### 3.2.3. Measurement in Cell Lysate

Vegetative *Dictyostelium* cells containing constitutive expression vectors (pDM317 hyg GFP or pDM317 diGFP) were put on LoFlo medium (Formedium) overnight, 10^8^ cells were pelleted, washed in 10 mM Na-K-hosphate buffer (pH = 6.5). Cells were lysed by incubation in 1 mL lysis buffer (50 mM Tris (pH 7.5), 50 mM NaCl, 5 mM DTT, 5 mM MgCl_2_, and 1% of a modified *Dictyostelium* protease inhibitor mix consisting of 2 µg/mL pepstatin (Carl Roth), 100 µg/mL N-tosyl-l-lysinc chlometyl ketone (Sigma), 80 µg/mL N-p-tosyl-l-arganine-methyl esther hcl (Sigma), 5 µg/mL leupeptin (Carl Roth), 0.1 mM PMSF (Carl Roth), 5 mM benzamidine (Sigma), 2 mM N-CBZ-Pro-ALA (Sigma) [41]) containing 1% triton for 30 min on ice.

Lysate was cleared from cell debris by centrifugation (10 min, 20,800× *g*, 4 °C). The protein concentrations were in the range of 5–30 mg/mL, as determined with the Bradford assay. The lysates were diluted 5–20-fold to a concentration corresponding to 3–5 molecules detected in the confocal spot (*N*) accordingly to the estimates reported by the ZEN acquisition software (Zeiss, Jena, Germany). To prepare solutions with the same *N,* the dilution factor was determined by measuring both the GFP and diGFP lysates. Subsequently, the lysates were mixed according to the percentage of diGFP as indicated. Lysates were kept on ice for a maximum of two hours.

All lysate samples were measured on a LSM710 ConfoCor 3 microscope (Zeiss, Jena, Germany), supplied with a 488 nm solid state laser, a triple dichroic (488/543/633) excitation filter, NFT 635 VIS dichroic mirror, BP 505-610 IR emission filter, Zeiss C-Apochromat 40x/1.2 NA water objective with coverslip thickness correction collar, and APD detector. The pinhole was set at 1 AU, and the laser power at 0.5%, as reported by the Zen software. Samples with a minimum volume of 100 µL were measured at 20 °C, in an 8-chambered coverslip (µ-Slide 8 Well Glass Bottom, Ibidi, Gräfelfing, Germany). Measurements were performed 30 µm above the glass surface, for 120–180 s in 1–3 spots per well. Rhodamine 110 (R110, *D* = 4.3 × 10^−10^ m^−2^ s^−1^ (Invitrogen, Breda, The Netherlands)) in water was used for calibration measurements.

#### 3.2.4. Measurement of Cells

*Dictyostelium* cells containing inducible expression vectors (pDM334hyg GFP, pDM334 diGFP, or pDM334 FKBP12 F36V) were starved for 3 h in Na-K-phosphate buffer (10 mM, pH = 6.5) at a cell density of 10^7^ cells per mL. DMSO was added to a final concentration of 2% *v*/*v* to impair cell movement, after which the cells were immediately transferred to 8-chambered coverslips (µ-Slide 8 Well Glass Bottom, Ibidi, Gräfelfing, Germany) and were allowed to settle down for 3–5 min. All cellular samples were measured on a TCS SP8 X SMD system (Leica Microsystems, Wetzlar, Germany), supplied with a super continuum laser (emitting a continuous spectrum from 470 to 670 nm), bandpass-adjustable spectral filters, a 63 × 1.20 NA water immersion objective with coverslip thickness correction collar, and a HyD internal hybrid detector, coupled to a PicoHarp 300 TCSPC module (PicoQuant, Berlin, Germany). The pinhole was set at 80 µm, the laser line at 488 nm with a pulsed frequency of 40 MHz, and the spectral filter at 495–545 nm. Raw intensity fluctuation data consisting of about 10^7^ photons were collected at 20 °C from single measurements. Generally, data were obtained from one measurement of 45 s per cell in one spot, in 10–20 cells per sample. Spots were selected in cells with low brightness, in areas with homogeneous distribution of fluorescence rather than nuclear area or large intracellular vesicles (Appendix A).

### 3.3. Data Analysis Procedure

#### 3.3.1. Fitting Software

The FCS and global FCS and PCH analyses were performed using the FFS data processor 2.6 software (SSTC, Department of Systems Analysis and Computer Modelling, Belarussian State University, Minsk, Belarus, www.sstcenter.com). This software offers the complete set of tools for the global analysis of FFS data. First the raw data, which is the sequence of photon arrival times stored in a binary form, are loaded in the program. This is followed by calculating a set of ACFs and PCDs, performing the non-linear constrained least-squares minimization, estimating of confidential intervals of fit parameters, displaying the analysis results is a user-friendly form, and finally storing both measured and analyzed data in databases. 

The software has a powerful algorithm to process raw data by splitting the measured data into parts and allowing for fully automated calculation of ACFs and PCDs at user defined binning times from each part. Weight factors for ACF are calculated as follows: the measurement (or part of the measurement) is split into a number of sub-parts, ACF is calculated from each sub-part and standard deviations of each point of resulted ACFs are calculated in a usual way. Weight factors for PCDs are calculated assuming the binomial distribution. The length of a PCD (a range of photon counts) is extended automatically by a factor of 2 each time the software counts more photons than allocated for the last channel. 

The non-linear constrained least-squares analysis is performed using the Levenberg–Marquardt optimization and reduced *χ*^2^ criterion. The fit parameters can be either constrained or fixed to the expected values. Confidential intervals are estimated using asymptotic standard errors (ASE) approach [42]. 

#### 3.3.2. Analysis of the Obtained Data

The obtained raw data were imported into the measurement database of the FFS data processor. Each measurement was divided into traces of 3–5 s, and for each trace one ACF and three PCDs were calculated. Each trace was subdivided by 10 for the ACF weight factors calculation. The ACF was calculated with 140 points and a time step of 1 × 10^−7^ s (cells) or 2 × 10^−7^ s (lysate). The lower binning times (1 × 10^−7^ s) and number of points in ACF (in comparison with conventionally used) were selected to ensure enough time length for the calculation of weights factors for ACF because of splitting the raw data in relatively short time traces. PCDs were calculated with 32 points as initial value and three different binning times, usually at 2 × 10^−5^, 5 × 10^−5^ and 1 × 10^−4^ s, depending on the total fluorescence intensity.

Next, the ACF curves from all traces of one measurement were analyzed globally using a free diffusion 3D Gaussian model (Equation (1)), with the parameters specified in Appendix A. Parameters *a*, *F_trip_*, *τ_trip_*, and *τ_diff_* were grouped between traces, and *a* was fixed to the value found for the R110 calibration dye. Confidence intervals (CI) with standard errors were calculated for *F_trip_*, *τ_trip_* and *τ_diff_*. In case of cellular data, the outcome of *F_trip_* and *τ_trip_* were averaged for all GFP (and diGFP) control samples of a single measurement day. These averaged values were fixed in the subsequent analyses of the experiment. 

Next, the PCD and ACF curves of all traces of one measurement were imported in a new worksheet and collectively analyzed using the BDGA method as described above (Equations (1)–(9)). For lysate data, parameter *a* was fixed to the value found in the control samples, while *F_trip_*, *τ_trip_*, and *τ_diff_* were fixed to the values found in the previous analyses of the ACF curves of that particular sample. For cells, *F_trip_* and *τ_trip_* were also based on the control samples. The background parameter *bg* was fixed to 0 because, for the first-order out-of-focus correction, the parameters *bg* and *Fc*1 are mutually correlated (Skakun et al., 2015) and one of them must be fixed, *G_inf_* was free, and *Fc*_1_ and *τ_dt_* were free but grouped for all traces together. The parameters *N* and *q* were grouped per trace, i.e., per one ACF and three PCD curves (see Appendix A for an illustration of the grouping). An example of the analysis output in the FFS data processor can be found in Appendix A.

All parameter estimates were imported into Microsoft Excel for the calculation of averages, standard deviations, statistics, and performing normalizations. In the case of lysate data, the apparent brightness *q* obtained from global analysis was recalculated into the true brightness *q_true_* by [43], using
(12)$$q_{\;true}=\;q_{\;0eff}/\left(1+Fc_{1}\right)$$

Each lysate sample was measured once or twice per experiment day and normalized towards GFP of that day. Because of the complex composition of the data (multiple experiment days, with variable multiple measurements per sample consisting of multiple traces), multiple options were explored to determine the best way to perform the statistics. Since measurements were performed during just a few days, and the number of measurements per day varied, data traces of different days were combined before averages and standard deviations were calculated. 

In the case of cellular data, averages were first calculated from the traces per cell, after which the averages of all cells (normalized towards GFP per day) from several days were calculated. Therefore, standard deviations here also account for the variation between cells. The standard t-test (unequal variances) was applied each time when the question of significance of the difference between obtained values was studied. 

## 4. Conclusions

Protein dimerization is abundant in nature [44] but studying the dimerization status of proteins in living cells is a challenge. In this paper we present a methodology to measure the oligomeric state of proteins using a standard confocal microscope including an FCS module. Diffusion and brightness information were combined using the BDGA method, exploiting all available information from the FCS data. We have developed a fully optimized analysis protocol by which monomeric and dimeric particles can be distinguished in cell extracts and in cells. 

We applied this methodology to analyze dimerization in living cells, as well as a mixture of GFP and diGFP in vitro. Compared to our previous global ACF and PCH experiments [26,27], changes were made regarding preparation of the samples, data acquisition, ACF and PCD curve calculation from raw data, global analysis of ACF and PCD curves, and further data processing. The changes regarding cellular sample preparation and data acquisition entailed (1) using starved rather than vegetative *Dictyostelium* cells, (2) adding 2% DMSO to the cells before the measurement, (3) selecting dim cells, (4) applying low laser power, (5) measuring for only 45 s per cell, and (6) using a sample for only 30 min. Regarding the raw data processing, measurements were separated into 3-s traces, from which ACF and PCD curves were then calculated as before. 

Regarding the global analysis of ACF and PCD curves (BDGA), the main adaptations entailed (1) discarding aberrant traces, (2) fixing parameters *a*, *F_trip_,*
*τ_trip_*, and *τ_diff_* (determined from only the ACF curves), rather than setting their values as initial guess, (3) grouping *N* and *q* per trace, rather than for all traces together, and (4) fixing *F_trip_* and *τ_trip_* for the cellular measurements with values as determined from the GFP sample. When processing the results, (1) the GFP sample rather than the R110 dye was used for normalization between measurement days, and (2) the apparent brightness rather than the true brightness (corrected for *Fc*_1_) was used for comparisons between cells. 

We have demonstrated that the ratio of monomer and dimer can be estimated with our methodology, by mixing the content of cells containing monomeric GFP and dimeric GFP. We have explained how we developed and optimized this method, and therefore several variants of the analysis as well as considerations regarding experimental setup and analysis were discussed. Next, we applied this method to distinguish GFP and diGFP in living cells, using *Dictyostelium discoideum* as a model organism. Finally, using the inducible dimerization domain FKBP12 [28,29], we have demonstrated that our BDGA method is well applicable to the in cellulo dimerization of GFP-tagged proteins. 

In our setup, dimerization of proteins with a monomeric GFP tag could be measured on a standard single-color, single-photon FCS microscope. Therefore, we consider this method to be very suitable for examining the dimerization status of any GFP-labeled cytosolic protein in living cells. Moreover, the presented methodology is in principle independent of the means of labeling and could therefore be applied to any fluorescent protein or synthetic dye. Besides cytosolic proteins, BDGA may also be applied to extracellular protein dimerization with no change in the used models. For non-mobile cells, regions at the cell surface may also be selected to study the oligomerization of membrane proteins, which should then be analyzed using 2D models rather than 3D (e.g., triplet-state with 2D diffusion model and the 2D Gaussian model for PSF profile). 

With the brightness and diffusion global analysis of FFS data, the abundant role of dimerization in nature can be further explored. 

## Figures and Tables

**Figure 1 ijms-22-07300-f001:**
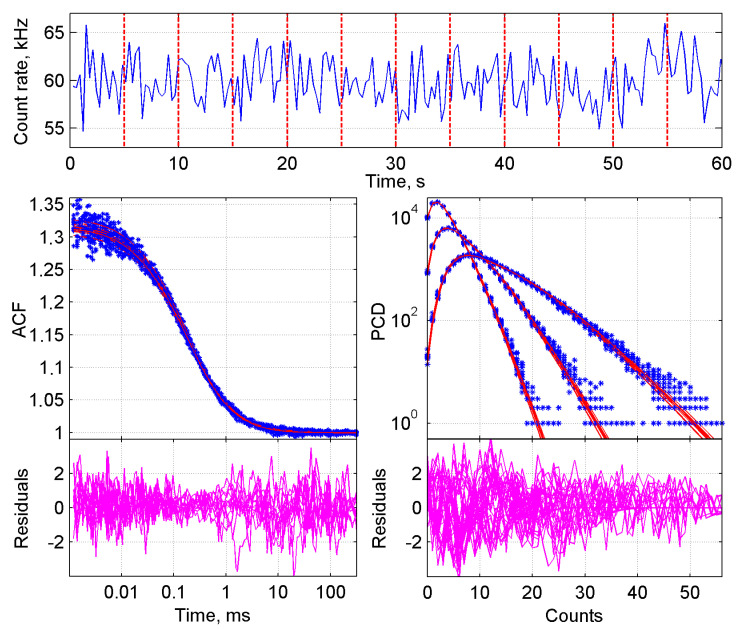
BDGA analysis of the ACF and PCD curves from a monomer-dimer ‘equilibrium’ in cell lysate. The sample consisted of 50% GFP and 50% diGFP, mimicking a stable equilibrium between two species. Top panel: raw FFS data showing photon counts over time, from which the ACF and PCD curves were calculated. The measurement was divided into twelve 5-s traces, as indicated by vertical lines. Bottom left: 1-component fit of all ACF curves, with residuals below. Bottom right: 1-component fit of all PCD curves, with residuals below. PCD curves were generated with three different time steps of 5 × 10^−5^, 1 × 10^−4^ and 2 × 10^−4^ s, respectively.

**Figure 2 ijms-22-07300-f002:**
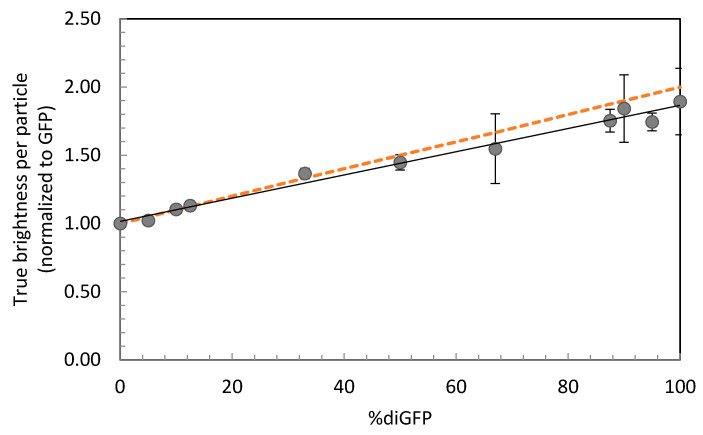
Average brightness of various GFP-diGFP mixes in cell lysate. Percentages of diGFP particles are relative to the total number of GFP and diGFP particles in the sample. Average true brightness is calculated as in Table 2, normalized to GFP per day. Error bars indicate standard deviations. The black line indicates the linear fit (R^2^ = 0.9837). The dashed line indicates the theoretical expected values, when a dimer would be twice as bright as a monomer.

**Figure 3 ijms-22-07300-f003:**
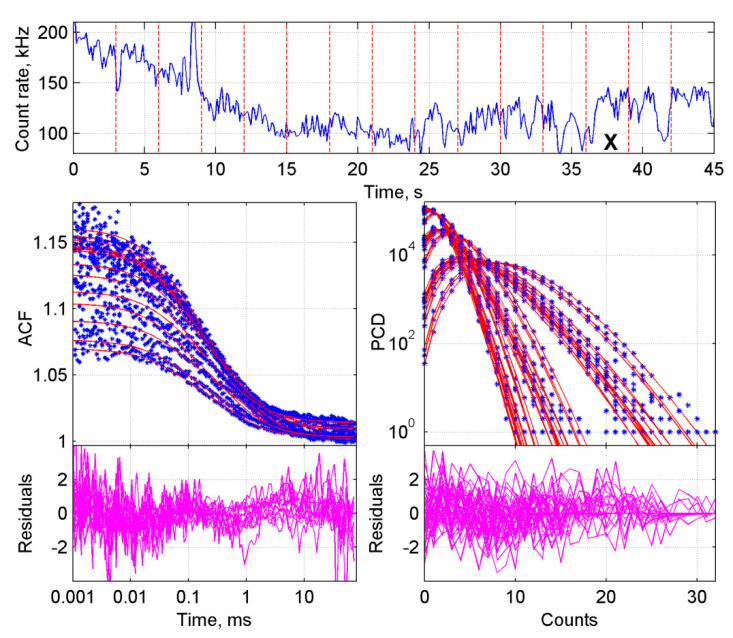
BDGA analysis of the ACF and PCD curves from GFP in living cells. The sample consisted of cells expressing GFP. Panels are explained in Figure 1. The measurement was divided into fifteen 3-s traces. Traces marked with an ‘X’ have been deleted based on visual inspection of the ACF curves. In the first 15 s, bleaching is taking place. The slight wave in the ACF residuals is expected because of internal movement of cellular compartments and fixation of the triplet state parameters. PCD curves were generated with three different time steps of 1 × 10^−5^, 2 × 10^−5^, and 5 × 10^−5^ s, respectively.

**Figure 4 ijms-22-07300-f004:**
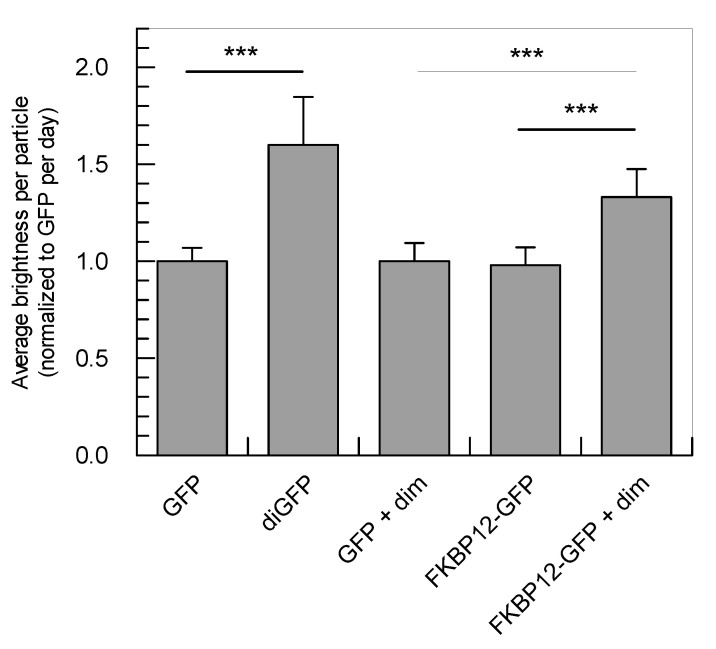
Average brightness calculated from analysis of multiple cells. The apparent brightness of several proteins was measured in living *Dictyostelium* cells and normalized to GFP per day. Averages were calculated from 24–35 cells per sample from 3 independent measurement days, as in Table 4. Error bars indicate standard deviations. FKBP12-GFP: GFP-linked dimerizing domain; +dim: cells have been incubated with 1 µM dimerizer for 3 h; *** *p* < 0.001.

**Figure 5 ijms-22-07300-f005:**
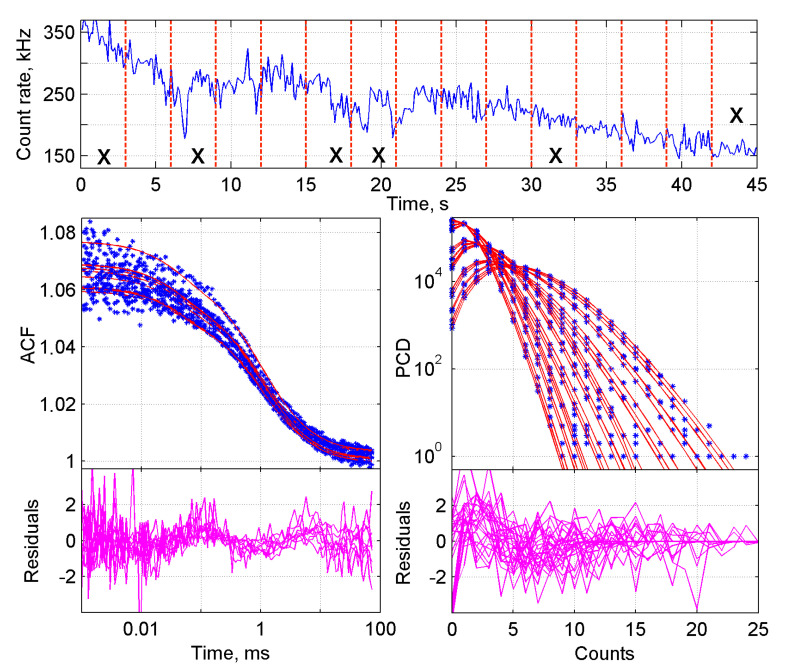
Results of BDGA analysis of the ACF and PCD curves from a monomer-dimer equilibrium of FKBP12 in living cells. The sample consisted of cells expressing FKBP12-GFP, supplemented with dimerizer. Panels are explained in Figure 1. The measurement was divided into fifteen 3-s traces. Traces marked with an ‘X’ have been deleted based on visual inspection of the ACF curves. PCD curves were generated with three different time steps of 5 × 10^−6^, 1 × 10^−5^, and 2 × 10^−5^ s, respectively.

**Table 1 ijms-22-07300-t001:** Analyses of representative lysate samples from one measurement day.

Sample	Analysis Method	*F_trip_* (×10^−2^)	*τ_trip_* (µs)	*τ_diff_*_1_ (µs)	*N* (±SD)	*q_true_* (×10^4^ cpms)	SD of *q_true_*	χ^2^
R110	1-component	9.10 ± 0.78	6.9 ± 1.2	35 ± 1	4.50 (±0.06)	4.04 ± 0.023	0.055	1.147
GFP	1-component	12.2 ± 0.90	35.1 ± 2.0	151 ± 4	3.88 (±0.09)	3.20 ± 0.020	0.096	1.05
diGFP	1-component	7.08 ± 0.43	31.0 ± 2.0	221 ± 3	3.63 (±0.06)	5.12 ± 0.023	0.090	1.233
50% GFP + 50% diGFP	1-component	8.97 ± 0.47	22.2 ± 2.5	191 ± 2	3.47 (±0.07)	4.88 ± 0.025	0.107	1.193
	2-component (*r* = 1.8)	8.07 ± 0.46	19.1 ± 2.4	*151 (τ_diff_* _1_ *); 221 (τ_diff_* _2_ *)*	2.00 (±0.18) (*N*_1_); 1.76 (±0.12) (*N*_2_)	*3.20 (q* _1_ *); 5.76 (q* _2_ *)*	-	1.068

Each measurement consists of 12 traces of 5 s. Deviations of all parameters except *N* are presented as confidence intervals, calculated as asymptotic standard errors (ASE), as reported by the software. *F_trip_*: triplet state fraction; *τ_trip_*: triplet state time; *τ_diff_*: diffusion parameter as reported by the software; *N*: number of particles in the confocal volume, with the standard deviation (SD) between traces indicated; *q_true_*: true brightness; χ^2^: value of the global fit criterion. ASEs represent uncertainties of estimated parameters obtained in analyses performed per one trace. For comparison we additionally calculated standard deviations of brightness between traces (SD of *q_true_*). Fixed values are indicated in italics (have been determined in the rows above). cpms: counts per molecule per second; *r* = *q*_2_/*q*_1_.

**Table 2 ijms-22-07300-t002:** Diffusion rate and brightness of various GFP-diGFP mixes in cell lysate.

diGFP%	*τ_diff_* (µs)	*D* (µm^2^ s^−1^)	*q_true_* (×10^4^ cpms)	*q_true_* Norm. to GFP
0	131 ± 21	97.3 ± 6.5	4.43 ± 1.22	1.00 ± 0.02
5	126 ± 32	98.3 ± 6.6	5.13 ± 1.19	1.02 ± 0.03
10	154 ± n.a.	99.0 ± n.a.	5.16 ± 1.60	1.10 ± 0.03
12.5	132 ± 37	94.1 ± 3.3	5.19 ± 1.15	1.13 ± 0.03
33.3	137 ± 32	87.7 ± 3.9	6.43 ± 1.42	1.37 ± 0.04
50	169 ± 31	74.0 ± 6.0	5.69 ± 1.62	1.45 ± 0.06
66.7	168 ± 37	78.4 ± 0.1	6.20 ± 1.84	1.55 ± 0.25
87.5	156 ± 37	76.8 ± 2.2	8.22 ± 2.02	1.75 ± 0.08
90	175 ± 46	69.5 ± 6.9	8.27 ± 2.88	1.84 ± 0.25
95	158 ± 36	75.7 ± 2.4	8.68 ± 1.70	1.74 ± 0.07
100	173 ± 40	71.4 ± 5.9	8.82 ± 2.77	1.89 ± 0.24

Percentages of diGFP particles are indicated, relative to the total number of GFP and diGFP particles in the sample. The presented values are mean ± standard deviation (SD) based on 42–126 traces from 2–5 measurement days. *τ_diff_*: diffusion parameter before any normalization; *D*: diffusion coefficient calculated from *τ_diff_*, corrected for the diffusion of R110 on each measurement day; *q_true_*: average true brightness without any correction for instrumental variation between days; *q_true_* norm. to GFP: true brightness normalized for the true brightness of the GFP sample per measurement day.

**Table 3 ijms-22-07300-t003:** Global analysis results of single cells.

Sample	*F_trip_*	*τ_trip_* (µs)	*τ_diff_* (µs)	*N* (range)	*q* (×10^4^ cpms)	SD of *q* (×10^4^ cpms)	# Traces	χ^2^
GFP (trip free)	0.241 ± 0.008	53.8 ± 2.8	381 ± 8	12.1 (8.5–19.9)	4.71 ± 0.06	0.55	14	1.197
GFP (trip fixed)	*0.178*	*40.0*	330 ± 2	11 (7.8–18.2)	5.18 ± 0.07	0.55	14	1.108
diGFP	*0.128*	*61.9*	841 ± 6	6.5 (4.9–7.9)	8.32 ± 0.10	1.04	13	1.079
FKBP12-GFP + dim	*0.178*	*40.0*	1018 ± 9	19 (16.5–21.7)	6.73 ± 0.18	0.79	9	1.015

Representative cells have been chosen from the same measurement day. Each measurement of 45 s was divided into 3-s traces. Deviations of all parameters except *N* are presented as confidence intervals. *F_trip_*, *τ_trip_*, and *τ_diff_*, see Table 1. *N*: number of particles in the confocal volume, lowest and highest values indicate the amount of bleaching; *q*: apparent brightness; # traces: number of traces from one measurement, on which the analysis was based; χ^2^: value of the global fit criterion. ASEs represent uncertainties of estimated parameters obtained in analyses performed per one trace. For comparison we additionally calculated standard deviations of brightness between traces (SD of *q*). +dim: 1 µM dimerizer was added to the cells for 3 h. Trip free: triplet state parameters were free in the ACF analysis. Fixed values are indicated in italics and have been determined by averaging multiple analyses with free triplet state parameters. The two GFP results were based on the same measurement but with different analysis settings.

**Table 4 ijms-22-07300-t004:** Induced dimerization of GFP in cells via the FKBP12 domain.

Sample	*τ_diff_* (µs)	*D* (µm^2^ s^−1^)	*q* (×10^4^ cpms)	*q* Normalized to GFP	# Cells
GFP	506 ± 115	16.7 ± 3.6	4.9 ± 0.6	1.00 ± 0.07	35
diGFP	871 ± 336	10.2 ± 2.4 *	7.8 ± 1.4	1.60 ± 0.25 *	27
GFP + dim	502 ± 133	17.1 ± 4.3	4.9 ± 0.6	1.00 ± 0.09	28
FKBP12-GFP	882 ± 269	9.8 ± 2.2	4.8 ± 0.7	0.98 ± 0.09	31
FKBP12-GFP + dim	1101 ± 336	7.8 ± 1.7 **	6.6 ± 0.9	1.33 ± 0.15 **	35

Addition of dimerizer increased the average brightness per particle when GFP was fused to the FKBP12 dimerizing domain, but not to GFP alone. The presented values are mean ± standard deviation (SD) based on 27–35 cells from 3 measurement days. Parameters are explained in Table 2 and Table 3. The diffusion coefficient *D* is calculated from *τ_diff_* and the diffusion rate of R110 on each measurement day. *q*: average apparent brightness without any correction for variation between days; *q* norm. to GFP: brightness normalized for the brightness of the GFP sample per measurement day. +dim: 1 uM dimerizer was added to the cells for 3 h.* *p* < 0.001 relative to GFP; ** *p* < 0.001 relative to GFP and FKBP12-GFP.

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
