# Peer review of "Combined FCS and PCH Analysis to Quantify Protein Dimerization in Living Cells"

_ijms, 2021, doi:10.3390/ijms22147300_

Round 1
Reviewer 1 Report
In this manuscript, the authors have nicely studied the quantification of protein dimerization in living cells using GFP proteins. Certainly, the protein-protein interaction and dimerization are one of the most important cell processes that need to be understood. Thus, the presented study will be of interest to a wide readership of the journal. I have the following comments that I would like to see in the revised version of the manuscript.
What is the author’s opinion about this - Is this method only applies to the GFP tagged protein dimerization or it can apply to the dye-labeled protein as well?
I would like to know if the present method can be applied to the protein outside of the cell or not – either extracellular oligomerization or cell surface proteins.
Please show the microscopic image in the figures that were used to analyze the data.
I would highly recommend giving a step-by-step process to follow in the supporting info file when other researchers want to follow this method – it will be very helpful.
Author Response
1- What is the author’s opinion about this - Is this method only applies to the GFP tagged protein dimerization or it can apply to the dye-labeled protein as well?
We thank the reviewer for this question. We have now addressed this in the discussion of the revised manuscript (Line 677 – 683). The method described in the paper can be applied to any dye-labeled protein. It can be well related to the category of general methods, which means it can be applied to study oligomerization of a wide range of substances.
2- I would like to know if the present method can be applied to the protein outside of the cell or not – either extracellular oligomerization or cell surface proteins.
We thank the reviewer for this constructive suggestion. The method can be well applied to the study of the extracellular oligomerization of proteins with no change in the used models (perhaps only the difference will be in selection of the region of interest). This has been added to the revised manuscript (Line 677 – 683).
3- Please show the microscopic image in the figures that were used to analyze the data.
We thank the reviewer for this suggestion, we have included this as Figure S1.
4- I would highly recommend giving a step-by-step process to follow in the supporting info file when other researchers want to follow this method – it will be very helpful.
We thank the reviewer for this recommendation. A brief overview of the analysis steps was added to the Supplementary Information.
Reviewer 2 Report
In this manuscript, Nederveen-Schippers and colleagues present the Brightness and Diffusion Global Analysis (BDGA), a novel tool to quantitively measure protein dimerization in living cells. The method is based on the combined analysis of the Fluorescence Correlation Spectroscopy (FCS) and the Photon Counting Histogram (PCH) and uses GFP-tagged proteins to correlate the average brightness per particle to the fraction of dimer present in each sample. As proof of principle, the authors used this methodology to measure dimerization of GFP, tandem-dimer GFP (diGFP), as well as ligand-induced dimerization of FKBP12-GFP.
Overall, this manuscript is understandable in most of its parts, at least for an audience of experts in this field. I think it is of interest for the audience of the International Journal of Molecular Science.
Minor points: The authors do not mention the meaning of the abbreviation “FCS” and “PCH” neither in the title or in the abstract, so the reader must go to through the paper to understand the subject of this paper. For a better understanding, please include their full name in extenso at least in the abstract.
Author Response
1- Minor points: The authors do not mention the meaning of the abbreviation “FCS” and “PCH” neither in the title or in the abstract, so the reader must go to through the paper to understand the subject of this paper. For a better understanding, please include their full name in extenso at least in the abstract.
We thank the reviewer for this remark. We have added an explanation of the abbreviations in the abstract.